# Testing the Effectiveness of *Curcuma longa* Leaf Extract on a Skin Equivalent Using a Pumpless Skin-on-a-Chip Model

**DOI:** 10.3390/ijms21113898

**Published:** 2020-05-29

**Authors:** Kyunghee Kim, Hye Mi Jeon, Kyung Chan Choi, Gun Yong Sung

**Affiliations:** 1Cooperative Course of Nano-Medical Device Engineering, Hallym University, Chuncheon 24252, Korea; facility.hee@gmail.com (K.K.); gpal1254@naver.com (H.M.J.); 2Integrative Materials Research Institute, Hallym University, Chuncheon 24252, Korea; 3Department of Pathology, Collage of Medicine, Hallym University, Chuncheon 24252, Korea; kcchoi@hallym.ac.kr; 4Major in Materials Science and Engineering, Hallym University, Chuncheon 24252, Korea

**Keywords:** pumpless skin-on-a-chip, human-skin equivalents, *Curcuma longa*, antiaging, microphysiological system

## Abstract

The in vitro tests in current research employ simple culture methods that fail to mimic the real human tissue. In this study, we report drug testing with a ‘pumpless skin-on-a-chip’ that mimics the structural and functional responses of human skin. This model is a skin equivalent constituting two layers of the skin, dermis and epidermis, developed using human primary fibroblasts and keratinocytes. Using the gravity flow device system, the medium was rotated at an angle of 15 degrees on both sides so as to circulate through the pumpless skin-on-a-chip microfluidic channel. This pumpless skin-on-a-chip is composed of upper and lower chips, and is manufactured using porous membranes so that medium can be diffused and supplied to the skin equivalent. Drug testing was performed using *Curcuma longa* leaf extract (CLLE), a natural product cosmetic ingredient, to evaluate the usefulness of the chip and the efficacy of the cosmetic ingredient. It was found that the skin barrier function of the skin epidermis layer is enhanced to exhibit antiaging effects. This result indicates that the pumpless skin-on-a-chip model can be potentially used not only in the cosmetics and pharmaceutical industries but also in clinical applications as an alternative to animal studies.

## 1. Introduction

Preclinical and clinical trials through animal experiments are essential to test the safety of novel drugs prior to their release. These trials are expensive, time-consuming, and require ethical clearances for the experiments; moreover, the observations often differ from the human responses.

Although understanding tissue formation, function, and response to pathology requires investigating the cells and tissues, their mechanical properties, and biochemical microenvironments, most of the current studies rely on conventional analysis of 2D cell cultures rather than reconstituting the in vivo cell microenvironment [1,2]. Since the 2D cell culture model does not accurately mimic the in vivo 3D environment, drug response data from these studies may be inaccurate. In order to overcome these limitations, collagen scaffolds have been developed to study 3D cell culture models [3,4,5,6,7]. The 3D culture models, exhibiting characteristics similar to the complex organisms, might present several advantages compared to 2D cell culture models, such as more accurate influence of cells on a micro environment, oxygen and nutrient gradients, increased interaction between a cell and its Extra-Cellular Medium (ECM), and more accurate depiction of cell proliferation [8,9,10,11,12]. We used pumpless skin-on-a-chip in our previous studies to provide nutrients to cells via the microfluidic channel and simulate the same various physiological functions, such as removing cell waste [3,4,5]. Since a previous study on three different type 1 collagens proved that the Rat Tail Collagen (RTC) is the most appropriate support for the 3D cultures [3], we used RTC to prepare a skin model in this study.

Here, a pumpless skin-on-a-chip model was used to test the efficacy of a drug from the natural product, *Curcuma longa* leaf extract (CLLE). *C. longa* is a perennial plant of the ginger family. The active constituents of *C. longa* are the flavonoid Curcuminoids, which are a mixture of curcumin (diferuloylmethane), monodexmethoxycurcumin and bisdesmethoxycurcumin. Curcumin makes up approximately 90% of the curcuminoid content in *C. longa*. Other constituents include sugars, proteins, and resins. The best-researched active constituent is curcumin, which comprises 0.3%–5.4% of raw *C. longa*. Commercially available curcumin used for research and for clinical trials (curcumin mix) contains ∼77% pure curcumin, 17% demethoxycurcumin and 3% bisdemethoxycurcumin [13,14,15]. Curcumin is traditionally used as a medicine for inflammation, stomach and liver diseases, diabetes, dermatitis, and arthritis. Further, anti-cancer, anti-oxidative and anti-inflammatory functions of curcumin have been recently reported [14,15,16]. However, such research is currently focused only on the rhizomes of *C. longa* but not extensively on by-products such as leaves and flowers. *C. longa* leaves are only partially used and their improper disposal also leads to environmental problems. Studies of CLLE are fewer than studies of roots and stems of curcuma; studies on medical applications of CLLE have been reported. Kim et al. reported that CLLE enhances skin immunity [17]. Lee et al. reported a novel extraction method for CLLE with low loss of active material [18]. Choi et al. reported that CLLE may promote the antioxidant activity of CLLE [19]. Using a balanced low pressure drying process, CLLE was made to promote dissolution of the bioactive agents. In this study, we investigated the anti-aging effect of CLLE by using pumpless skin-on-a-chip as a test tool.

## 2. Results

### 2.1. Examination of Tissue Morphology with H&E Staining

The skin equivalent 3D culture system was prepared with pumpless skin-on-a-chip using 6.12 mg RTC, and treated with under air exposure in the differentiation stage with 0, 50, and 250 μg/mL CLLE. The formation of the stratum corneum, upon CLLE treatment for 3, 5, and 7 days, was confirmed by H & E staining (Figure 1). Samples treated without CLLE were considered as negative controls for treated groups. A 3-day 50 μg/mL CLLE treatment was the best condition for stratum corneum formation. Treatment with 50 μg/mL and 250 μg/mL CLLE for 5 days resulted in a well-differentiated stratum corneum layer compared to the negative controls. When CLLE was treated for 7 days, it was confirmed that keratinization was well observed in 50 μg/mL CLLE. However, the group treated with 0 and 250 μg/mL CLLE for 7days showed relatively poor keratinization.

### 2.2. Measurement of the Thickness of the Stratum Corneum

The thickness of the stratum corneum and spinosum layer was quantitated using H&E staining (Figure 2). While the spinosum layer gradually became thicker with longer treatments, the stratum corneum remained unaltered in negative controls. While 50 μg/mL CLLE treatment resulted in gradual thickening of the spinosum layer, 250 μg/mL CLLE treatment resulted in irregular thickening. Further, the stratum corneum became gradually thicker with 50 μg/mL CLLE but remained unaltered with 250 μg/mL CLLE. By the 7^th^ day, an average of about 22 μm thickness is considered optimal for stratum corneum.

### 2.3. qRT-PCR Results for Quantitative Analysis

Healthy and young skin is well formed with the function of the skin barrier. As a marker of skin barrier function, we used gene expressions of filaggrin, involucrin, Keratin 10, and Laminin alpha-5. These genes are known to increase gene expression when the skin barrier is well formed, and the results are confirmed through qPCR.

Expression of filaggrin gene remained unaltered by the 3^rd^ day in the negative control and in 50 μg/mL CLLE-treated samples, but upregulated by more than two-fold in 250 μg/ml CLLE-treated samples (Figure 3a). By the 5^th^ day, it was upregulated in CLLE-treated samples. By the 7^th^ day, the highest expression was observed in the 50 μg/mL CLLE-treated sample. 

Involucrin, upregulated during differentiation, forms a hydrophobic cornified cell envelope, which acts as a physical barrier, on the keratinocyte outer membrane. Involucrin gene was upregulated by the 3^rd^ day of CLLE treatment. Although upregulated by the 5^th^ and 7^th^ days in samples treated with 50 μg/mL CLLE when compared to those with 250 μg/mL CLLE, from the 7^th^ day, a downregulation was observed in the sample treated with 250 μg/mL CLLE, which led to an expression level lower than that of the negative control (Figure 3b). 

Keratin 10 gene is downregulated in the CLLE-treated samples (compared to the negative control) by the 3^rd^ day, but upregulated by the 5^th^ and 7^th^ days (Figure 3c). 

Although Laminin alpha-5 was upregulated in 50 and 250 μg/mL CLLE-treated samples until the 5^th^ day (Figure 3d), similar levels were observed in 0 and 250 μg/mL CLLE-treated samples by the 7^th^ day. Fifty micrograms per milliliter CLLE treatment led to downregulation in long-term culture until day 7. Filaggrin, involucrin, and laminin alpha-5 showed negative control level or lower level expression. 

### 2.4. Immunohistochemistry (IHC) Result of Staining Analysis

The production levels were further compared based on their proportion (0%–10%; 1 points, >10–50%; 2 points, >50%–75%; 3 points, >75%; 4 points) and intensity (minimal, 1 points; mild, 2 points; moderate, 3 points; severe, 4 points) [20], between the samples treated with different concentrations of CLLE and for different durations (Figure 8). 

The expression level of fibronectin in Figure 4 was expressed over a larger area in Figure 4d–f than in Figure 4a–c, and was most strongly expressed in Figure 4g–i. Each expression level was graphically compared to Figure 8 via the score method. The results of the filaggrin IHC staining in Figure 5 show the highest expression level of those treated with CLLE for 3 days (Figure 5a–c), except for the experimental group (Figure 5h) treated with 50 μg/mL CLLE for 7 days, the expression level tended to decrease as the treatment period of CLLE increased.

Involucrin IHC results in Figure 6 showed that the expression level increased as the CLLE treatment period became longer. Figure 6g–i shows stronger expression levels than Figure 6a–c. 

Figure 8c, which is relatively compared by the score method, shows a rapid increase in the expression of Involucrin, especially in the experimental group treated with 50 μg/mL CLLE.

The Keratin 10 IHC results in Figure 7 show that overall Keratin 10 expression is displayed at similar levels. A graph (Figure 8d) quantifying and comparing the results shows that the group treated with CLLE has a relatively higher value than that of the negative control not treated with CLLE, and there is no difference in value according to the treatment period.

## 3. Discussion

In order to evaluate the usefulness of the chips and the efficacy of the cosmetic ingredients, a drug test was performed using the natural product cosmetic ingredient, *Curcuma longa* leaf extract (CLLE). H & E staining was performed to confirm the formation of skin equivalents and reaction to drugs. Using the staining result, the tissue morphology was examined to measure the thickness of the stratum corneum. The epidermis acts as a first line of defense through its different layers that harbor keratinized keratinocytes, which are responsible for maintaining healthy and stable skin. Therefore, since treatment with 50 μg/mL CLLE for 3, 5 and 7 days resulted in optimal skin differentiation and keratinization, this concentration is considered as the most appropriate treatment. Further, treatment CLLE for 7 days or longer was unproductive; Skin equivalent without CLLE treatment for 7 days showed normal tissue formation. Treatment with 250 μg/mL CLLE was ineffective and could be toxic to the tissues, due to a five-fold higher concentration than the optimum (50 μg/mL). In this paper, it was difficult to distinguish the precise layers in H&E staining; the stratum corneum and the lucidum layers together are referred to as the stratum corneum, while the basal, spinosum, and granulosum layers together are referred to as the spinosum layer.

H&E staining showed that the stratum corneum and the spinosum layers became thicker as the CLLE concentration and the culture period increased (Figure 1). Thus, it is thought that CLLE contributes to activate differentiation of keratinocytes, which forms the stratum corneum layer. Increased synthesis of the constituent proteins and lipids leads to the optimal thickness (~16 μm) of spinosum layer.

In the formation of the epidermis layer, CLLE contributes to the division and differentiation of keratinocytes in the basal layer. Keratinocytes differentiate to form corneocytes. Keratinization and shedding results in the overproduction of various proteins and lipids, resulting in relatively large amounts of lipid accumulation between keratinocytes. Therefore, thickness was observed upon H&E staining in 7 days samples treated with 50 and 250 μg/mL CLLE. Thickening confirmed that the formation of the stratum corneum is independent of hyperkeratosis and that the development of the stratum corneum is closely related to the formation of the skin barrier. Therefore, CLLE could preserve skin moisture and nutrients and regulate skin protection from external factors such as bacteria and viruses.

The thickness of the stratum corneum and spinosum layer was quantitated using H&E staining result in Figure 1 (Figure 2). Figure 2 shows that 50 μg/mL CLLE treatment is effective in gradually increasing the thickness of the stratum corneum and spinosum layer during the air exposure period. Specifically, the thickness of the stratum corneum was between 16 and 22 μm and did not increase beyond 22 μm even by increasing treatment concentration and duration. The maximum limit that can be achieved is expected to be about 22 μm. The thickness of the stratum corneum of the epidermis layer in the reconstructed human epidermal model currently marketed was reported at 15–32 μm [7,21,22]. Since our results (about 22 μm thickness) are comparable to the normal range, it is considered that the stratum corneum layer has been well formed in this study. Since the skin is exfoliated from the stratum corneum layer surface by the proliferated basal layer and there is a continuous epidermal homeostasis, it can be considered that the thickness of the stratum corneum layer is maintained at a constant level. 

The epidermis of skin is composed of the basal layer, stratum spinosum layer, and stratum corneum layer that harbor keratinocytes at different stages of differentiation. Epidermal cell proliferation, differentiation, and death occur sequentially, and each stage is characterized by the expression of specific protein-encoding genes. Expression levels of these genes were quantified as a function of CLLE concentration and duration, using qRT-PCR.

Filaggrin, a filament-associated protein, facilitates binding of keratinocyte’s Keratin fiber to epithelial tissue. It forms a mesh-like structure in the stratum corneum to moisturize the skin barrier and promotes developing into hydroscopic amino acids, which constitutes an element of a natural moisturizing factor [12,23]. Figure 3a result indicates that 50 μg/mL CLLE treatment leads to filaggrin overproduction in the epidermal granulosum layer and thus could form the stratum corneum having an optimal moisture-retaining function. Involucrin, upregulated during differentiation, forms a hydrophobic cornified cell envelope, which acts as a physical barrier on the keratinocyte outer membrane. Figure 3b indicates that involucrin may be overproduced in 50 μg/mL CLLE-treated samples, and a more robust epidermis layer is formed. Keratin is responsible for a variety of phenotypes: a) overproduced when a cornified cell envelope is formed, b) promotes expression of differentiation biomarkers, and c) maintains epidermal homeostasis [24,25]. Figure 3c indicates that a more cornified cell envelope may be formed when treated with 50 μg/mL CLLE and the expression levels increased by the duration of treatment. Laminin alpha-5 plays an important role in the biological activity of the basement membrane, in cell differentiation, migration, adhesion, cell phenotype, and survival. The Laminin alpha-5 molecule, with its three short arms, forms a framework for capturing other cells [26,27,28]. Filaggrin, involucrin, and laminin alpha-5 showed negative control level or lower level expression. Therefore, the optimal concentration of CLLE was found to be 50 μg/mL and the optimal culture period was 5 days. As shown in the H & E staining images, when CLLE was treated at 50 μg/mL, high expression of filaggrin, involucrin, laminin alpha-5 and keratin 10 genes, which are genes related to skin barrier strengthening and recovery, was observed. Thus, it is established that CLLE will help restore the skin barrier. The skin barrier is composed of keratinocytes containing the natural moisturizing factor in the epidermis layer with the intercellular lipid filled in between [23]. The skin barrier can be damaged by innate factors, environmental stress, cosmetics including surfactant components, and aging. Further, breaking of the skin barrier in allergen-sensitive skin may lead to dermatitis.

Since changes at the gene level were not directly linked to changes at the protein level, the protein levels within the skin equivalent were estimated in addition to the gene expression levels. Production for four proteins was directly investigated by IHC (Figure 4, Figure 5, Figure 6 and Figure 7).

In order to confirm that the skin equivalent is well formed, we observed the main forming protein of the dermis layer, fibronectin. As shown in Figure 4, fibronectin, an abundant human skin glycoprotein, mediates the contact between cellular elements and collagen [29,30,31]. In a 14 days 3D culture of fibroblasts, to form the dermis, fibronectin was found to be affected by the culture period rather than CLLE treatment (Figure 4). Since the dermis layer was well formed, we inferred that long-term culturing is ideal for dermis formation (Figure 8a).

The main protein components in the granulosum layer are keratin and filaggrin, which constitute 80% to 90% of the mass of the epidermis [32]. Filaggrin is an important component of the cornified envelope of the outer layer of the epidermis. Loss of filaggrin leads to loss of water (exocytosis) in the labile stratum corneum [12,33]. Relative proportions and production intensity of flaggrin in the stratum corneum layer were evaluated through the scoring method (Figure 5 and Figure 8b). In particular, 50 μ/mL CLLE treatment led to relatively higher levels of filaggrin and consequently optimal water levels in the stratum corneum (Figure 4b,e,h). This is consistent with the qRT-PCR result (Figure 3). 

Involucrin, a precursor of the cross-linked outer shell protein in human stratum corneum, is a sensitive marker for epithelial keratosis [34,35]. Involucrin expression is initiated in the early spinosum layer and maintained in the granulosum [36,37]. Involucrin production gradually increased upon longer CLLE treatment, with a rapid increase in 50 μg/mL CLLE-treated samples (Figure 6 and Figure 8c). This result is consistent with the gene expression profiles (Figure 3). Therefore, CLLE treatment not only changes the gene level but also changes protein production, indicating that changes in the expression level of involucrin protein actually occurred at the skin equivalent. This indicates the formation of a water-insoluble cornified cell envelope [38], upon CLLE treatment, for strengthening the physical barrier of the skin. 

Conversion to abandoned cells that contributed to differentiation in undifferentiated basal keratinocytes is accompanied by structural changes in integral ECM receptors, accompanied by a decrease in adhesion of the substrate layer and changes in the stratum corneum synthesis program [39]. Keratin 1 and keratin 10, the major secondary differentiation-specific keratins of interfollicular epidermis, are expressed by the suprabasal epidermis and any other stratified squamous epithelia that become orthokeratinized [39,40,41]. Expression of keratin 1 and keratin 10 appears to inhibit cell proliferation and cells moving up into the suprabasal layers become post-mitotic, and progressively more terminally differentiated as they continue their journey upwards the epidermal surface. Keratin 10 is a marker of hyperdifferentiation and differentiation of the epidermis and is expressed throughout the epidermis. Keratin 10 deficiency can lead to changes in the permeability barrier function of the skin [41,42,43]. Keratin 10 is overproduced in the CLLE-treated sample compared to that in the negative control (Figure 7 and Figure 8d). This result is consistent with the thickness of the stratum corneum (Figure 2), although the change depending on the treatment period is not remarkable. This could be because the post-mitotic process of keratinocytes has progressed sufficiently from the third day of CLLE treatment (third day of skin equivalent air exposure culture) in achieving an appropriate thickness for the stratum corneum, suggesting a longer duration for these cultures. This indicates that in the longer CLLE treatment periods, stratum corneum homeostasis is maintained but no side effects, such as hyperkeratosis, were detected.

## 4. Materials and Methods

### 4.1. Fabrication of a Pumpless Microfluidic Chip and Gravity Flow System

The skin-on-a-chip fabrication process is based on soft lithography of polydimethylsiloxane (PDMS) (base:agent = 10:1), as described in the previous reports [3,4]. We previously reported a gravity flow system that overcomes the existing limitation of the 2D cell culture static environment and provides a medium for the microfluidic channel [3]. The medium was designed to be delivered to the 3D cell culture chamber through a microfluidic channel and membrane in the chip medium chamber (Figure 9). The medium chamber was designed as a cylinder 8 mm in diameter at the center of the chip and the medium chamber was made with a structure connected to both sides of three cylinders that are 8 mm in diameter. The culture medium is supplied to the chamber via a patterned lower PDMS chip with a microfluidic channel with a width of 200 μm and a height of 150 μm and polyester membrane (Corning Inc., Wiesbaden, Germany) (Figure 9b). The gravity flow system could control the angle and time (Figure 9e). The principle of operation is that the medium flows through the microfluidic channel of the pumpless microfluidic chip along the slope of the gravity flow system device and is supplied to the 3D culture scaffold via the membrane at the bottom of the culture chamber. This system is connected to a PC that controls the motor and a chip holder that operates the repetitive fluctuation.

### 4.2. Construction of a 3D Skin Equivalent Model

Two human primary cells are employed: fibroblasts (5.0 × 10^5^, Biosolution Co., Ltd., Seoul, Rep. of Korea), which are dermal cells and keratinocytes (1.0 × 10^6^, Biosolution Co., Ltd., Seoul, Rep. of Korea). The fibroblasts and keratinocytes were cultured in Fibroblast Growth Medium (FGM, Lonza Inc., Basel, Switzerland) and Keratinocyte culture medium (KGM-Gold, SingleQuots, Lonza Inc., Basel, Switzerland), respectively. Fibroblasts were seeded on 6.12 mg RTC (Corning Inc., New York, USA), cultured for 4–5 days, seeded with keratinocytes, and stabilized for 3–5 days (EGF-1 10 ng/mL, Hydrocortisone 0.4 μg/mL, Insulin 5 μg/mL, Transferrin 5 μg/mL, and DMEM/Ham’s F12). The cells were cultured using 2 × 10^−11^ M 3,3,5-triiodo-l-thionine sodium salt, 10^−10^ M Cholera toxin, 10% *v*/*v* FBS and 1% penicillin/streptomycin; the medium was replaced every 24 h. The above process is schematically shown in Figure 10. CLLE powder (Shebah Biotech Inc., Chuncheon, Rep. of Korea) was mixed with E-media to a final concentration of 50 μg/mL and 250 μg/mL. 

### 4.3. Real-time Quantitative Analysis

For quantitative Real-time Polymerase Chain Reaction (qRT-PCR) analysis, mRNA was extracted using TRIzol method and quantified in SpectraMax M2 Microplate Readers (Molecular Devices Inc., San Jose, CA, USA). The cDNA was synthesized using amfiRivert cDNA synthesis Platinum Master Mix (GenDEPOT, Barker, TX, USA). The qPCR was run using the LightCycler^®^ 480 SYBR Green I Master (Roche, Basel, Switzerland) in LightCycler^®^ 480 Instrument II (Roche Basel, Switzerland). The following primer pairs were used; for human 18s rRNA gene 5’-GGCGCCCCCTCGATGCTCTTAG-3’ and 5’-GCTCGGGCCTGCTTTGAACACTCT-3’; for human Filaggrin gene, 5’-GGAGTCACGTGGCAGTCCTCACA-3’ and 5’-GGTGTCTAAACCCGGATTCACC-3’; for human Involucrin gene, 5’-CCGCAAATGAAACAGCCAACTCC-3’ and 5’-GGATTCCTCATGCTGTTCCCAG-3’, for human Laminin 5 gene, 5’-GGAACTTCCGGCATACGGAGA-3’ and 5’-GGACAGGCACAGCTCCACATT-3’, for human Keratin 10 gene, 5’-CCGGAGATGGTGGCCTTCTCTCT-3’ and 5’-GGCCTGATGTGAGTTGCCATGCT-3’.

### 4.4. H&E Staining and Immunohistochemistry

For hematoxylin and eosin (H&E) staining, tissues were embedded in paraffin and sectioned at 4 μm thickness. After deparaffinization with xylene and hydration with alcohol wash, sections were stained with hematoxylin solution (Sigma-Aldrich Co., St. Louis, MO, USA) for 30 s. After washing with tap water, sections were stained with eosin solution (Sigma-Aldrich Co., MO, USA) for 1 min. After washing with distilled water, sections were dehydrated with an alcohol wash and unstained with xylene. The stained tissues were observed in an inverted light microscope (Olympus, DP73, Japan).

For immunohistochemistry (IHC) staining, the slides were incubated with primary antibodies specific to Fibronectin (ab2413), Involucrin (ab53112), Cytokeratin (ab76318), Laminin 5 (ab14509), and Filaggrin (ab81468), separately overnight at 4 °C and processed using Benchmark XT Auto-stainer System (Roche), following the manufacturer’s instructions. All antibodies were purchased from Abcam Inc, Cambridge, UK and used at 1:200 dilution. 

## 5. Conclusions

In this study, a human skin equivalent model consisting of the epidermal and dermal layers was fabricated using a microfluidic platform pumpless skin-on-a-chip. In order to confirm the possibility of utilizing it as a drug testing model, CLLE, a natural anti-aging skin cosmetic was used to confirm its anti-aging effect on the skin. H&E staining was used to observe the structure of human skin equivalent epidermis, dermis and endothelium. The changes in gene expression levels of filaggrin, involucrin, laminin alpha-5, and keratin 10, as quantified by qRT-PCR analysis, were compared with IHC staining. In this experiment, CLLE at concentrations of 0, 50, and 250 μg/mL were used for 3, 5, and 7 days to study the effects of treatment period and drug concentration. Fifty micrograms per milliliter CLLE treatment showed the best recovery effect of the skin and thus could be employed in prevention of aging. Therefore, the in vitro microfluidic platform, pumpless skin-on-a-chip, could co-culture fibroblast cells and keratinocyte cells to establish a human skin equivalent model by mimicking the structure, function, and physiology of human skin. It is possible to evaluate the efficacy of natural drugs using this model. The pumpless skin-on-a-chip has potential clinical applications and can be applied in the pharmaceutical and cosmetics industries for animal experimentation.

## Figures and Tables

**Figure 1 ijms-21-03898-f001:**
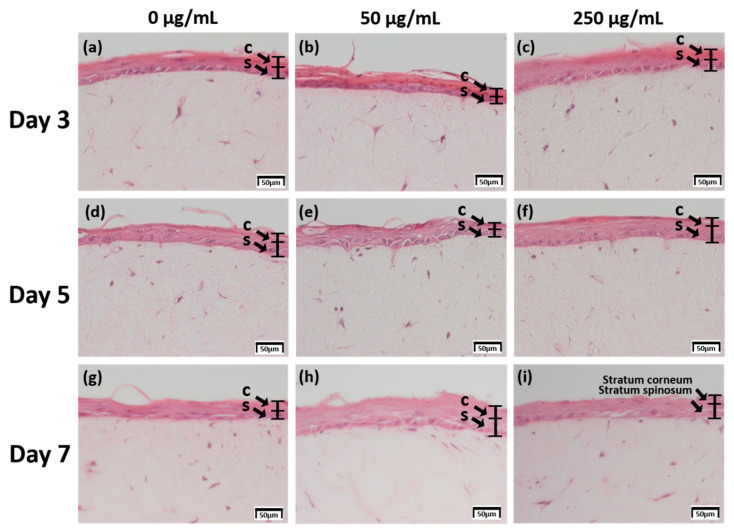
H & E staining images of 3D cell cultured sample for 3 days air exposure using culture media-treated CLLE concentrations of (**a**) 0 μg/mL, (**b**) 50 μg/mL, and (**c**) 250 μg/mL; for 5 days air exposure using culture media-treated CLLE concentrations of (**d**) 0 μg/mL, (**e**) 50 μg/mL, and (**f**) 250 μg/mL; and for 7 days air exposure using culture media-treated CLLE concentration of (**g**) 0 μg/mL, (**h**) 50 μg/mL, and (**i**) 250 μg/mL. (C; Stratum corneum, S; Stratum spinosum)

**Figure 2 ijms-21-03898-f002:**
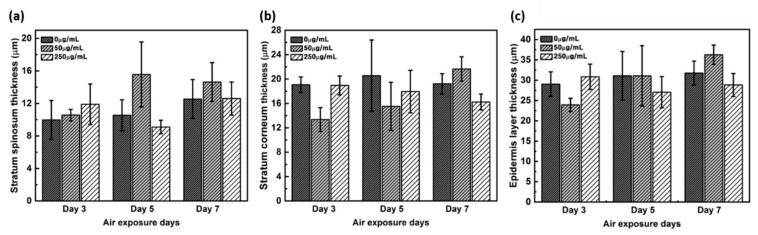
Thickness of (**a**) stratum corneum layer, (**b**) spinosum layer, and (**c**) total thickness of the stratum corneum layer and spinosum layer measured from H&E stained images as a function of air exposure days and concentration of CLLE.

**Figure 3 ijms-21-03898-f003:**
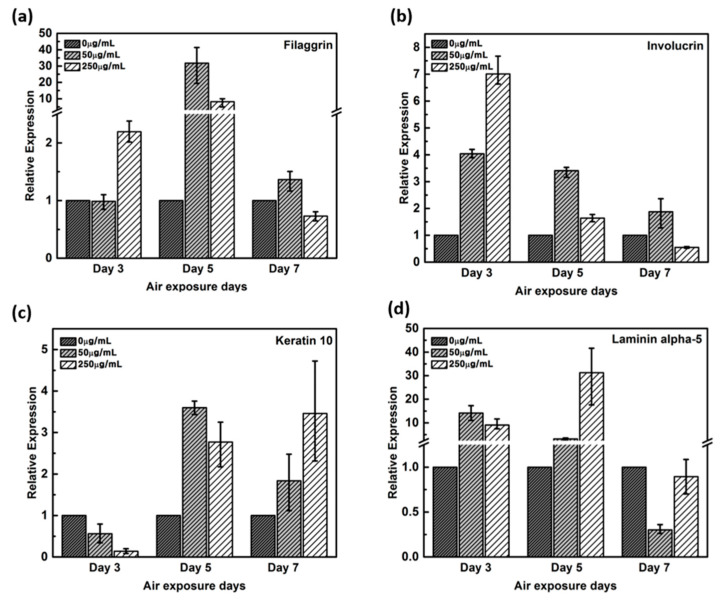
Relative gene expression of (**a**) filaggrin, (**b**) involucrin, (**c**) keratin 10, (**d**) laminin alpha-5 with varying CLLE-treated concentrations and period by real time qPCR. It quantified relative to negative control in each air exposure period on the basis of negative control.

**Figure 4 ijms-21-03898-f004:**
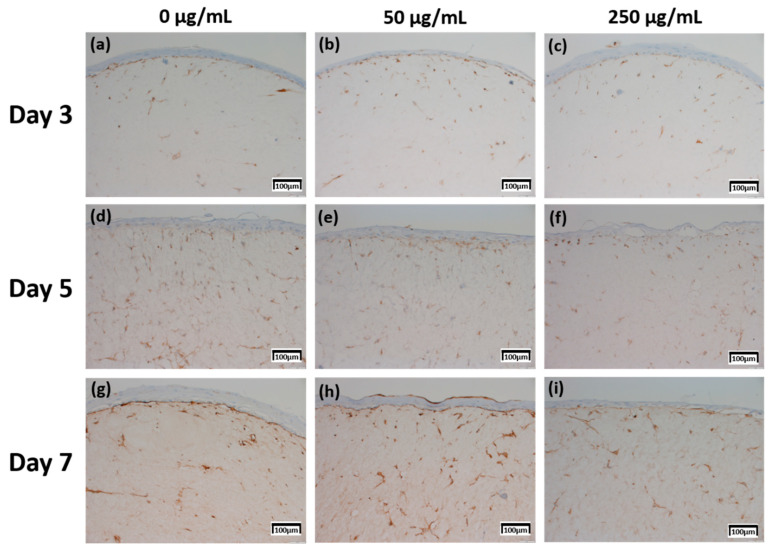
Immunohistochemistry stained images for fibronectin of 3D cell cultured sample for 3 days air exposure using culture media-treated CLLE concentrations of (**a**) 0 μg/mL, (**b**) 50 μg/mL, and (**c**) 250 μg/mL; for 5 days air exposure using culture media-treated CLLE concentrations of (**d**) 0 μg/mL, (**e**) 50 μg/mL, and (**f**) 250 μg/mL; and for 7 days air exposure using culture media-treated CLLE concentrations of (**g**) 0 μg/mL, (**h**) 50 μg/mL, and (**i**) 250 μg/mL.

**Figure 5 ijms-21-03898-f005:**
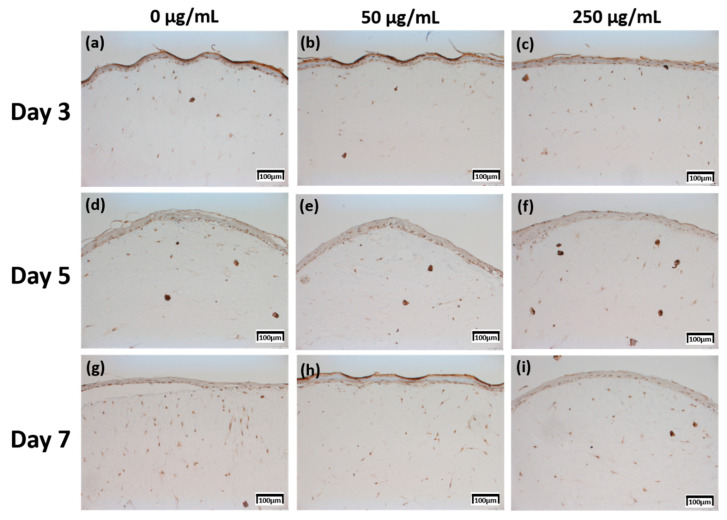
Immunohistochemistry stained images for filaggrin of 3D cell cultured sample for 3 days air exposure using culture media-treated CLLE concentrations of (**a**) 0 μg/mL, (**b**) 50 μg/mL, and (**c**) 250 μg/mL; for 5 days air exposure using culture media-treated CLLE concentrations of (**d**) 0 μg/mL, (**e**) 50 μg/mL, and (**f**) 250 μg/mL; and for 7 days air exposure using culture media-treated CLLE concentrations of (**g**) 0 μg/mL, (**h**) 50 μg/mL, and (**i**) 250 μg/mL.

**Figure 6 ijms-21-03898-f006:**
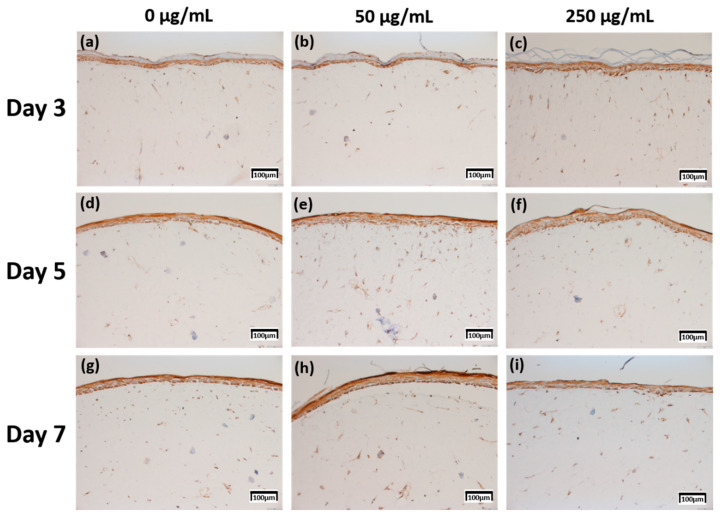
Immunohistochemistry stained images for involucrin of 3D cell cultured sample for 3 days air exposure using culture media-treated CLLE concentrations of (**a**) 0 μg/mL, (**b**) 50 μg/mL, and (**c**) 250 μg/mL; for 5 days air exposure using culture media-treated CLLE concentrations of (**d**) 0 μg/mL, (**e**) 50 μg/mL, and (**f**) 250 μg/mL; and for 7 days air exposure using culture media-treated CLLE concentrations of (**g**) 0 μg/mL, (**h**) 50 μg/mL, and (**i**) 250 μg/mL.

**Figure 7 ijms-21-03898-f007:**
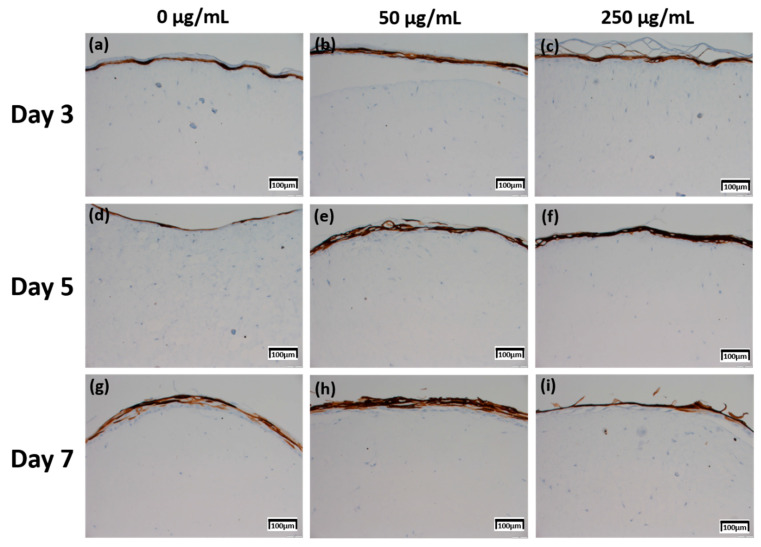
Immunohistochemistry stained images for keratin 10 of 3D cell cultured sample for 3 days air exposure using culture media-treated CLLE concentrations of (**a**) 0 μg/mL, (**b**) 50 μg/mL, and (**c**) 250 μg/mL; for 5 days air exposure using culture media-treated CLLE concentrations of (**d**) 0 μg/mL, (**e**) 50 μg/mL, and (**f**) 250 μg/mL; and for 7 days air exposure using culture media-treated CLLE concentrations of (**g**) 0 μg/mL, (**h**) 50 μg/mL, and (**i**) 250 μg/mL.

**Figure 8 ijms-21-03898-f008:**
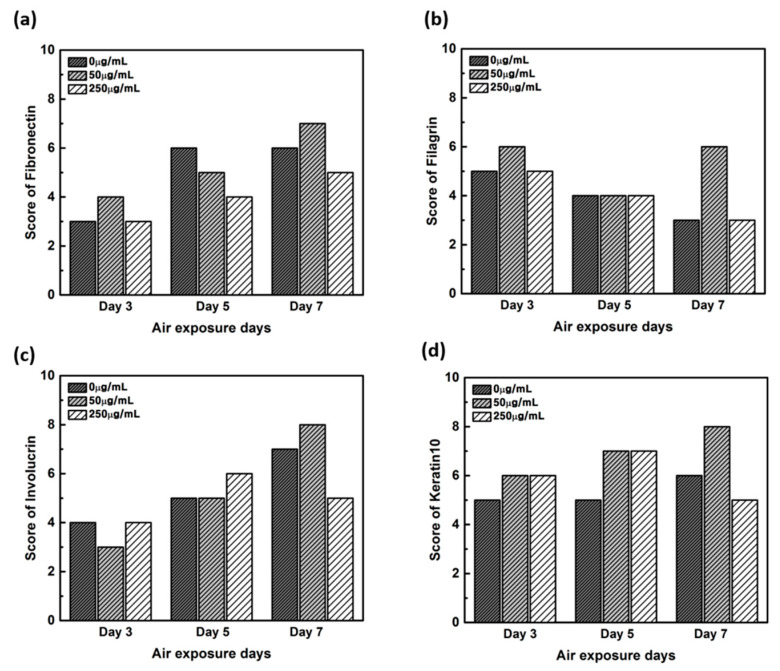
Quantitative analysis of the protein expression using the scoring method on the immunohistochemistry stained images. (**a**) Fibronectin, (**b**) filaggrin, (**c**) involucrin, and (**d**) keratin.

**Figure 9 ijms-21-03898-f009:**
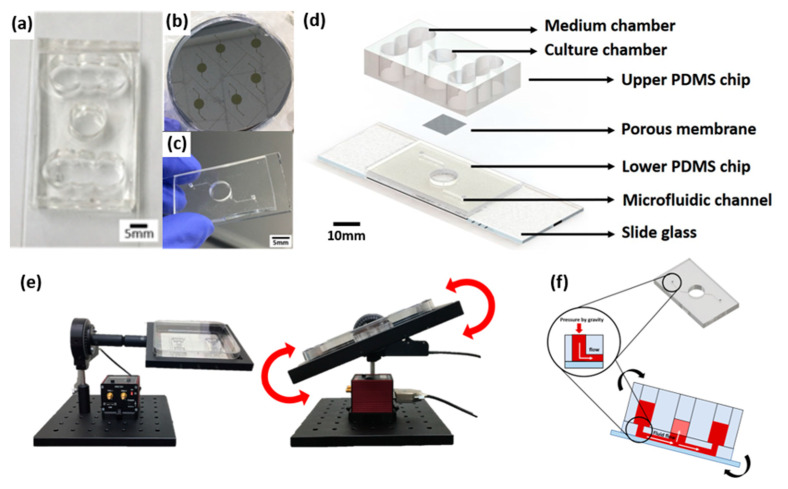
A schematic diagram of a microfluidic skin-on-a-chip before assembly and a gravity flow system. (**a**) Upper PDMS chip. (**b**) Microfluidic channels patterned mask mold. (**c**) The lower chip with the microfluidic channel patterned. (**d**) Configuration diagram of pumpless microfluidic skin-on-a-chip. (**e**) View of the front and side of the gravity flow system. It works by shaking both sides at 15° degrees. A 15° degree tilt causes the medium to circulate through the microfluidic channel. (**f**) A schematic diagram of a microfluidic skin-on-a-chip before assembly.

**Figure 10 ijms-21-03898-f010:**
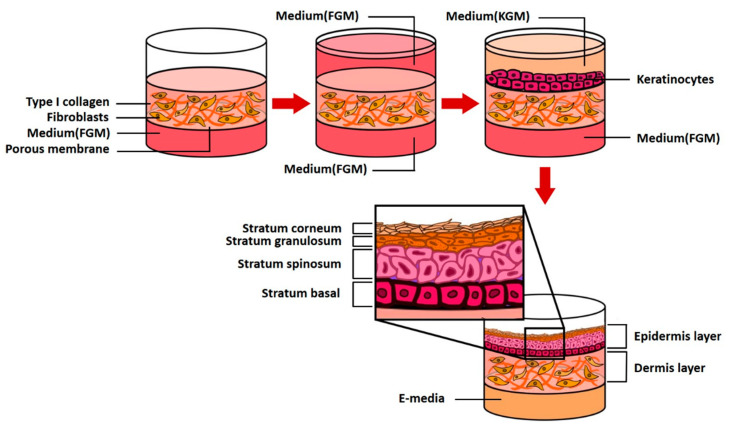
A schematic diagram of 3D skin model formation process in a pumpless microfluidic skin-on-a-chip.

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
