# Peer review of "Testing the Effectiveness of *Curcuma longa* Leaf Extract on a Skin Equivalent Using a Pumpless Skin-on-a-Chip Model"

_ijms, 2020, doi:10.3390/ijms21113898_

Round 1
Reviewer 1 Report
Good written paper.
Author Response
Thank you for your review.
Reviewer 2 Report
Literature:
Curcuma longa is a commonly used spice throughout the world, and has been shown to exhibit anti inflammatory, antimicrobial, antioxidant, and anti-neoplastic properties. Growing evidence shows that an active component of Curcuma longa, may be used medically to treat a variety of dermatologic diseases. Therapeutic properties have been studied in the past:
- Effects of Turmeric (Curcuma longa) on Skin Health: A Systematic Review of the Clinical Evidence.
https://www.ncbi.nlm.nih.gov/pubmed/27213821
- Potential of Curcumin in Skin Disorders
https://www.ncbi.nlm.nih.gov/pmc/articles/PMC6770633/
- Curcuma Longa As Medicinal Herb In The Treatment Of Diabet- Ic Complications
https://www.ncbi.nlm.nih.gov/pubmed/29624265
Pumpless microfluidic platform for drug testing on human skin equivalents
https://www.ncbi.nlm.nih.gov/pmc/articles/PMC4305008/
Development of 3D skin-equivalent in a pump-less microfluidic chip
https://www.sciencedirect.com/science/article/abs/pii/S1226086X17306275
However, further studies will be essential to better evaluate the efficacy and the mechanisms involved.
Highlights from the manuscript:
The authors investigated the anti-aging effect of CLLE by using pumpless skin-on-a-chip as a test tool.
Important observations in the article:
- “While the spinosum layer gradually became thicker with longer treatments, the stratum corneum remained unaltered in negative controls. While 50 μg/mL CLLE treatment resulted in gradual thickening of the spinosum layer, 250 μg/mL CLLE treatment resulted in irregular thickening. Further, the stratum corneum became gradually thicker with 50 μg/mL CLLE but remained unaltered with 250 μg/mL CLLE.”
- “treatment with 50 μg/mL CLLE for 3, 5 and 7 days resulted in optimal skin differentiation and keratinization, this concentration is considered as the most appropriate treatment.”
- “CLLE contributes to activate differentiation of keratinocytes”
Observations and comments:
The authors make a very good introduction and the language used is excellent.
a). Some histological measurements on the images, showing these thicknesses, would be very useful for readers, especially in Figures 1, 4, 5, 6, and 7. Arrows to indicate the layers in these images would be very useful for readers who are less versed.
b). In figure 2, I believe that the two panels should be combined (or a third panel should be inserted). I believe that will show the dynamics between stratum corneum layer and spinosum layer – over time.
c). I don't understand what is the purpose of the this text at the end of Figure 1 ?: “This is a figure, Schemes follow the same formatting. If there are multiple panels, they should be listed as: (a) Description of what is contained in the first panel; (b) Description of what is contained in the second panel. Figures should be placed in the main text near to the first time they are cited. A caption on a single line should be centered.”
d). Please clarify the phrase: “Further, treatment for 7 days or longer was unproductive; the stratum corneum in the skin equivalent did not.”
e). The two phrases are not too clear: “Upon 7 days of treatment, skin stratification was observed with 50 μg/mL CLLE. However, with 0 and 250 μg/mL CLLE, increased keratinization was not observed from corresponding 5 day-treatment groups.”
f). A brief description of the four genes and the diseases associated with them would be useful to the reader. The reader needs to know why the expression of this specific set of genes was followed.
To add more value and originality, I think the above points should be considered. However, the manuscript can be also published with a few minor modifications (eg observation c).).
Author Response
Dear Reviewer :
We appreciate the comment. We agreed with this reviewer’s opinion and the manuscript was revised including Figure 1 and Figure 2.
a). Some histological measurements on the images, showing these thicknesses, would be very useful for readers, especially in Figures 1, 4, 5, 6, and 7. Arrows to indicate the layers in these images would be very useful for readers who are less versed.
==> Arrows are marked to each layers in Fig.1
b). In figure 2, I believe that the two panels should be combined (or a third panel should be inserted). I believe that will show the dynamics between stratum corneum layer and spinosum layer – over time.
==> In order to show the change in the overall thickness of the stratum corneum layer and spinosum layer, we added the results showing the total thickness of the corneum layer and spinosum layer into Figure 2.
c). I don't understand what is the purpose of the this text at the end of Figure 1 ?: “This is a figure, Schemes follow the same formatting. If there are multiple panels, they should be listed as: (a) Description of what is contained in the first panel; (b) Description of what is contained in the second panel. Figures should be placed in the main text near to the first time they are cited. A caption on a single line should be centered.”
==> The content of the template was not removed properly. It has nothing to do with this paper. This part was removed.
d). Please clarify the phrase: “Further, treatment for 7 days or longer was unproductive; the stratum corneum in the skin equivalent did not.”
==>
In the above results (figures 1, 3, 4, 5, 6, 7, and 8), when CLLE was treated for more than 7 days, the results of qPCR showed that the gene expression of filaggrin, involucrin, and Laminin alpha-5 decreased compared to the group treated with 5 days. In the IHC results, except for the 50 μg / mL CLLE-treated group on 7 days, the 250 μg / mL-treated group on Day 7 showed inefficient results compared to the without CLLE-treated group. This was the explanation. The contents have been revised as follows for more understanding. “Further, treatment CLLE for 7 days or longer was unproductive; Skin equivalent without CLLE treatment for 7 days showed normal tissue formation. Treatment with 250 μg/mL CLLE was ineffective and could be toxic to the tissues, due to a five-fold higher concentration than the optimum (50 μg/mL).”
e). The two phrases are not too clear: “Upon 7 days of treatment, skin stratification was observed with 50 μg/mL CLLE. However, with 0 and 250 μg/mL CLLE, increased keratinization was not observed from corresponding 5 day-treatment groups.”
==>
The sentence explaining the difference in keratinization by concentration of CLLE treatment in the 7th day sample was not clear, so the sentence was corrected as follows. “When CLLE was treated for 7 days, it was confirmed that keratinization was well observed in 50 μg / mL CLLE. However, the group treated with 0 and 250 μg / mL CLLE for 7days showed relatively poor keratinization.”
f). A brief description of the four genes and the diseases associated with them would be useful to the reader. The reader needs to know why the expression of this specific set of genes was followed.
==> It seems that there is a lack of explanation for gene expression, so I added this explanation to the result section. “Healthy and young skin is well formed with the function of the skin barrier. As a marker of skin barrier function, we used gene expressions of filaggrin, involucrin, Keratin 10, and Laminin alpha-5. These genes are genes known to increase gene expression when the skin barrier is well formed, and the results are confirmed through qPCR.”
Reviewer 3 Report
The paper entitled "Testing the Effectiveness of Curcuma longa Leaf
Extract on a Skin Equivalent Using a Pumpless Skin-on-a-chip Model" described the fabrication of a human skin equivalent model consisting of the epidermal and dermal layers using a microfluidic platform pumpless skin-on-a-chip in order to get a much closer idea of the effect of substances using a 3D model when compared to the use of 2D cell cultures. The idea is good and the results are interesting, nevertheless as the authors also want to test the Effectiveness of Curcuma longa Leaf Extract, at this point the paper can be greatly improved. There is no information on the composition of this Curcuma longa Leaf Extract. If the idea was to see only of the device has a good response the authors could only use a positive known control substance in the pure form. Therefore as the topic of the Curcuma longa Leaf Extract is also relevant for the study, I think that a better characterization of the extract use should be given in the manuscript.
Author Response
Thank you for your comment. In this manuscript, we focussed on the usefulness of a pumpless skin-on-a-chip model for testing the effectiveness of Curcuma longa leaf extract. Characteristics of the extract have been well-known as we summarized in introduction section through Ref. 13 ~ 19.
Round 2
Reviewer 3 Report
I Agree you the authors that they provided references in the introductions about the potential composition of leaf extracts, nevertheless as they are leaf extracts, a natural product, the variability of the initial composition of the leafs together with the variability that can be introduced in the extraction process, we cannot really know what is present in the extract studied in this work, unless the extract used in this work has been previously characterized and the results are published our available to the reader.